# WEIGHTED LINE GRAPH CONVOLUTIONAL NETWORKS

## ABSTRACT

Line graphs have shown to be effective in improving feature learning in graph neural networks. Line graphs can encode topology information of their original graphs and provide a complementary representational perspective. In this work, we show that the encoded information in line graphs is biased. To overcome this issue, we propose a weighted line graph that corrects biases in line graphs by assigning normalized weights to edges. Based on our weighted line graphs, we develop a weighted line graph convolution layer that takes advantage of line graph structures for better feature learning. In particular, it performs message passing operations on both the original graph and its corresponding weighted line graph. To address efficiency issues in line graph neural networks, we propose to use an incidence matrix to accurately compute the adjacency matrix of the weighted line graph, leading to dramatic reductions in computational resource usage. Experimental results on both real and simulated datasets demonstrate the effectiveness and efficiency of our proposed methods.

## 1 INTRODUCTION

Graph neural networks (Gori et al., 2005; Scarselli et al., 2009; Hamilton et al., 2017) have shown to be competent in solving challenging tasks in the field of network embedding. Many tasks have been significantly advanced by graph deep learning methods such as node classification tasks (Kipf & Welling, 2017; Veličković et al., 2017; Gao et al., 2018), graph classification tasks (Ying et al., 2018; Zhang et al., 2018), link prediction tasks (Zhang & Chen, 2018; Zhou et al., 2019), and community detection tasks (Chen et al., 2019). Currently, most graph neural networks capture the relationships among nodes through message passing operations. Recently, some works (Chen et al., 2019) use extra graph structures such as line graphs to enhance message passing operations in graph neural networks from different graph perspectives. A line graph is a graph that is derived from an original graph to represent connectivity between edges in the original graph. Since line graphs can encode the topology information, message passing operations on line graphs can enhance network embeddings in graph neural networks. However, graph neural networks that leverage line graph structures need to deal with two challenging issues; those are bias and inefficiency. Topology information in original graphs is encoded in line graphs but in a biased way. In particular, node features are either overstated or understated depending on their degrees. Besides, line graphs can be much bigger graphs than original graphs depending on the graph density. Message passing operations of graph neural networks on line graphs lead to significant use of computational resources.

In this work, we propose to construct a weighted line graph that can correct biases in encoded topology information of line graphs. To this end, we assign each edge in a line graph a normalized weight such that each node in the line graph has a weighted degree of 2. In this weighted line graph, the dynamics of node features are the same as those in its original graph. Based on our weighted line graph, we propose a weighted line graph convolution layer (WLGCL) that performs a message passing operation on both original graph structures and weighted line graph structures. To address inefficiency issues existing in graph neural networks that use line graph structures, we further propose to implement our WLGCL via an incidence matrix, which can dramatically reduce the usage of computational resources. Based on our WLGCL, we build a family of weighted line graph convolutional networks (WLGCNs). We evaluate our methods on graph classification tasks and show that WLGCNs consistently outperform previous state-of-the-art models. Experiments on simulated data demonstrate the efficiency advantage of our implementation.

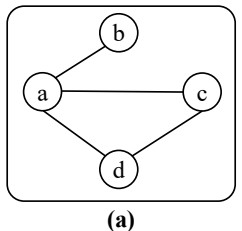 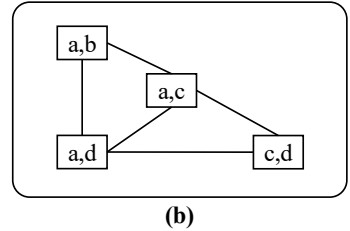 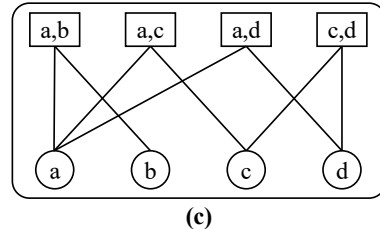

Figure 1: Illustrations of an graph (a), its corresponding line graph (b), and its incidence graph (c).

## 2 BACKGROUND AND RELATED WORK

In graph theory, a line graph is a graph derived from an undirected graph. It represents the connectivity among edges in the original graph. Given a graph $\mathbb{G}$, the corresponding line graph $L(\mathbb{G})$ is constructed by using edges in $\mathbb{G}$ as vertices in $L(\mathbb{G})$. Two nodes in $L(\mathbb{G})$ are adjacent if they share a common end node in the graph $\mathbb{G}$ (Golumbic, 2004). Note that the edges $(a, b)$ and $(b, a)$ in an undirected graph $\mathbb{G}$ correspond to the same vertex in the line graph $L(\mathbb{G})$. The Whitney graph isomorphism theorem (Thatte, 2005) stated that a line graph has a one-to-one correspondence to its original graph. This theorem guarantees that the line graph can encode the topology information in the original graph. Recently, some works (Monti et al., 2018; Chen et al., 2019; Bandyopadhyay et al., 2019; Jiang et al., 2019) proposes to use the line graph structure to enhance the message passing operations in graph neural networks. Since the line graph can encode the topology information, the message passing on the line graph can enhance the network embeddings in graph neural networks. In graph neural networks that use line graph structures, features are passed and transformed in both the original graph structures and the line graph structures, thereby leading to better feature learnings and performances.

## 3 WEIGHTED LINE GRAPH CONVOLUTIONAL NETWORKS

In this work, we propose the weighted line graph to address the bias in the line graph when encoding graph topology information. Based on our weighted line graph, we propose the weighted line graph convolution layer (WLGCL) for better feature learning by leveraging line graph structures. Besides, graph neural networks using line graphs consume excessive computational resources. To solve the inefficiency issue, we propose to use the incidence matrix to implement the WLGCL, which can dramatically reduce the usage of computational resources.

### 3.1 BENEFIT AND BIAS OF LINE GRAPH REPRESENTATIONS

In this section, we describe the benefit and bias of using line graph representations.

**Benefit** In message-passing operations, edges are usually given equal importance and edge features are not well explored. This can constrain the capacity of GNNs, especially on graphs with edge features. In the chemistry domain, a compound can be converted into a graph, where atoms are nodes and chemical bonds are edges. On such kinds of graphs, edges have different properties and thus different importance. However, message-passing operations underestimate the importance of edges. To address this issue, the line graph structure can be used to leverage edge features and different edge importance (Jiang et al., 2019; Chen et al., 2019; Zhu et al., 2019). The line graph, by its nature, enables graph neural networks to encode and propagate edge features in the graph. The line graph neural networks that take advantage of line graph structures have shown to be promising on graph-related tasks (Chen et al., 2019; Xiong et al., 2019; Yao et al., 2019). By encoding node and edge features simultaneously, line graph neural networks enhance the feature learning on graphs.

**Bias** According to the Whitney graph isomorphism theorem, the line graph $L(\mathbb{G})$ encodes the topology information of the original graph $\mathbb{G}$, but the dynamics and topology of $\mathbb{G}$ are not correctly represented in $L(\mathbb{G})$ (Evans & Lambiotte, 2009). As described in the previous section, each edge in the graph $\mathbb{G}$ corresponds to a vertex in the line graph $L(\mathbb{G})$. The features of each edge contain features of its two end nodes. A vertex with a degree $d$ in the original graph $\mathbb{G}$ will generate

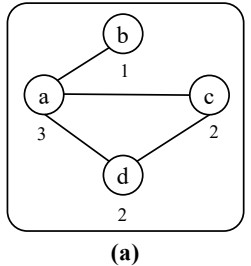 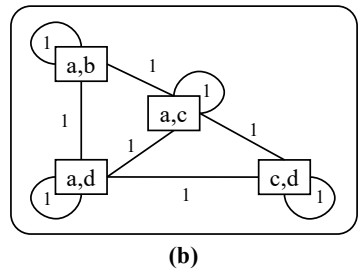 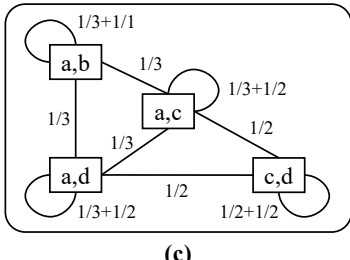

(a)           (b)           (c)

Figure 2: An illustration of a graph (a), its corresponding line graph (b), and its weighted line graph (c). Here, we consider a graph with 4 nodes and 4 edges as illustrated in (a). The numbers show the node degrees in the graph. In figure (b), a line graph is constructed with self-loops. Each node corresponds to an edge in the original graph. In the regular line graph, the weight of each edge is 1. Figure (c) illustrates the weighted line graph constructed as described in Section 3.2. The weight of each edge is assigned as defined in Eq. (1).

$d \times (d-1)/2$ edges in the line graph $L(\mathbb{G})$. The message passing frequency of this node's features will change from $O(d)$ in the original graph to $O(d^2)$ in the line graph. From this point, the line graph encodes the topology information in the original graph but in a biased way. In the original graph, a node's features will be passed to $d$ neighbors. But in the corresponding line graph, the information will be passed to $d \times (d-1)/2$ nodes. The topology structure in the line graph $L(\mathbb{G})$ will overstate the importance of features for nodes with high degrees in the graph. On the contrary, the nodes with smaller degrees will be relatively understated, thereby leading to biased topology information encoded in the line graph. Note that popular adjacency matrix normalization methods (Kipf & Welling, 2017; Veličković et al., 2017; Gao & Ji, 2019; Gong & Cheng, 2019) cannot address this issue.

## 3.2 WEIGHTED LINE GRAPH

In the previous section, we show that the line graph $L(\mathbb{G})$ constructed from the original graph $\mathbb{G}$ encodes biased graph topology information. To address this issue, we propose to construct a weighted line graph that assigns normalized weights to edges. In a regular line graph $L(\mathbb{G})$, each edge is assigned an equal weight of 1, thereby leading to a biased encoding of the graph topology information. To correct the bias, we need to normalize edge weights in the line graph.

Considering each edge in $\mathbb{G}$ has two ends, it is intuitive to normalize the weighted degree of the corresponding node in $L(\mathbb{G})$ to 2. To this end, the weight for an edge in the adjacency matrix $\boldsymbol{F}$ of $L(\mathbb{G})$ is computed as:

$$F_{(a,b),(b,c)} = \begin{cases} \frac{1}{D_b} & \text{if } a \neq c \\ \frac{1}{D_b} + \frac{1}{D_a}, & \text{if } a = c \end{cases} \tag{1}$$

where $a$, $b$, and $c$ are nodes in the graph $\mathbb{G}$, $(a,b)$ and $(b,c)$ are edges in the graph $\mathbb{G}$ that are connected by the node $b$. $D_b$ is the degree of the node $b$ in the graph $\mathbb{G}$. To facilitate the message passing operation, we add self-loops on the weighted line graph $WL(\mathbb{G})$. The weights for self-loop edges computed by the second case consider the fact that they are self-connected by both ends. Figure 2 illustrates an example of a graph and its corresponding weighted line graph.

**Theorem 1.** *Given the edge weights in the weighted line graph $WL(\mathbb{G})$ defined by Eq. (1), the weighted degree for a node $(a,b)$ in $WL(\mathbb{G})$ is 2.*

The proof of Theorem 1 is provided in the supplementary material. By constructing the weighted line graph with normalized edge weights defined in Eq. (1), each node $(a,b)$ has a weighted degree of 2. Given a node $a$ with a degree of $d$, it has $d$ related edges in $\mathbb{G}$ and $d$ related nodes in $L(\mathbb{G})$. The message passing frequency of node $a$'s features in the weighted line graph $WL(\mathbb{G})$ is $\sum_{i=1}^{d} 2 = O(d)$, which is consistent with that in the original graph $\mathbb{G}$. Thus, the weighted line graph encodes the topology information of the original graph in an unbiased way.

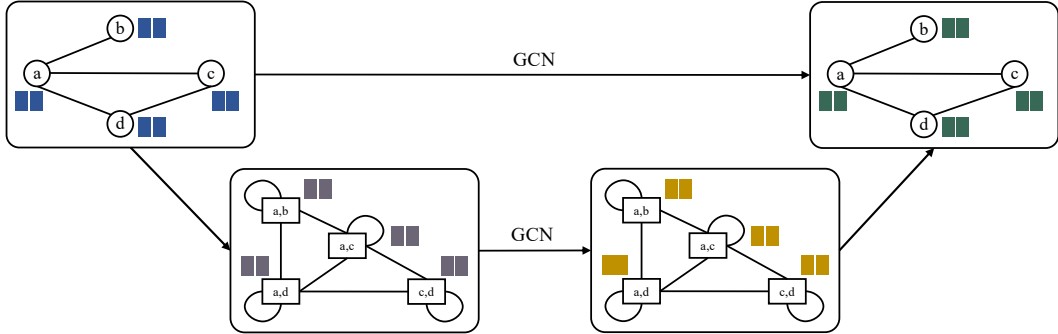

Figure 3: An illustration of our proposed weighted line graph convolution layer. We consider an input graph with 4 nodes and each node contains 2 features. Based on the input graph, we firstly construct the weighted line graph with features as described in Section 3.2. Then we apply two GCN layers on the original graph and the weighted line graph, respectively. The edge features in the line graph are transformed back into node features and combined with features in the original graph.

### 3.3 WEIGHTED LINE GRAPH CONVOLUTION LAYER

In this section, we propose the weighted line graph convolution layer (WLGCL) that leverages our proposed weighted line graph for feature representations learnings. In this layer, node features are passed and aggregated in both the original graph $\mathbb{G}$ and the corresponding weighted line graph $WL(\mathbb{G})$.

Suppose an undirected attributed graph $\mathbb{G}$ has $N$ nodes and $E$ edges. Each node and each edge in the graph contains $C_n$ and $C_e$ features, respectively. In the layer $\ell$, an adjacency matrix $\boldsymbol{A}^{(\ell)} \in \mathbb{R}^{N \times N}$, a node feature matrix $\boldsymbol{X}^{(\ell)} \in \mathbb{R}^{N \times C_n}$, and a edge feature matrix $\boldsymbol{Y}^{(\ell)} \in \mathbb{R}^{E \times C_e}$ are used to represent the graph connectivity, node features, and edge features, respectively. Here, we construct the adjacency matrix $\boldsymbol{F}^{(\ell)} \in \mathbb{R}^{E \times E}$ of the corresponding weighted line graph. The layer-wise propagation rule of the weighted line graph convolution layer $\ell$ is defined as:

$$\hat{\boldsymbol{Y}}^{(\ell)} = \boldsymbol{F}^{(\ell)} \boldsymbol{Y}^{(\ell)}, \qquad\qquad \in \mathbb{R}^{E \times C_e} \qquad (2)$$

$$\boldsymbol{K}_L^{(\ell)} = \boldsymbol{B}^{(\ell)} \hat{\boldsymbol{Y}}^{(\ell)}, \qquad\qquad \in \mathbb{R}^{N \times C_e} \qquad (3)$$

$$\boldsymbol{K}^{(\ell)} = \boldsymbol{A}^{(\ell)} \boldsymbol{X}^{(\ell)}, \qquad\qquad \in \mathbb{R}^{N \times C_n} \qquad (4)$$

$$\boldsymbol{X}^{(\ell+1)} = \boldsymbol{K}^{(\ell)} \boldsymbol{W}^{(\ell)} + \boldsymbol{K}_L^{(\ell)} \boldsymbol{W}_L^{(\ell)}, \qquad\qquad \in \mathbb{R}^{N \times C'} \qquad (5)$$

where $\boldsymbol{W}^{(\ell)} \in \mathbb{R}^{C_n \times C'}$ and $\boldsymbol{W}_L^{(\ell)} \in \mathbb{R}^{C_e \times C'}$ are matrices of trainable parameters. $\boldsymbol{B}^{(\ell)} \in \mathbb{R}^{N \times E}$ is the incidence matrix of the graph $\mathbb{G}$ that shows the connectivity between nodes and edges.

To enable message passing on the line graph $L(\mathbb{G})$, each edge in the graph $\mathbb{G}$ needs to have features. However, edge features are not available on some graphs. To address this issue, we can compute features for an edge $(a, b)$ by summing up features of its two end nodes: $Y_{(a,b)}^{(\ell)} = X_a^{(\ell)} + X_b^{(\ell)}$. Here, we use the summation operation to ensure the permutation invariant property in this layer. Then, we perform message passing and aggregation on the line graph in Eq. (2). With updated edges features, Eq. (3) generates new nodes features with edge features $\boldsymbol{Y}^{(\ell)}$. Eq. (4) performs feature passing and aggregation on the graph $G$, which leads to aggregated nodes features $\boldsymbol{K}^{(\ell)}$. In Eq. (5), aggregated features from the graph $\mathbb{G}$ and the line graph $L(\mathbb{G})$ are transformed and combined, which produces the output feature matrix $\boldsymbol{X}^{(\ell+1)}$. Note that we can apply popular adjacency matrix normalization methods (Kipf & Welling, 2017) on the adjacency matrix $\boldsymbol{A}^{(\ell)}$, the line graph adjacency matrix $\boldsymbol{F}^{(\ell)}$, and the incidence matrix $\boldsymbol{B}^{(\ell)}$.

In the WLGCL, we use the line graph structure as a complement to the original graph structure, thereby leading to enhanced feature learnings. Here, we use a simple feature aggregation method as used in GCN (Kipf & Welling, 2017). Other complicated and advanced feature aggregation methods such as GAT (Veličković et al., 2017) can be easily applied by changing Eq. (2) and Eq. (4) accordingly. Figure 3 provides an illustration of our WLGCL.

Figure 4: An illustration of the weighted line graph convolution network. The input graph is an undirected attributed graph. Each node in the graph contains two features. Here, we use a GCN layer to produce low-dimensional continuous feature representations. In each of the following two blocks, we use a layer and a layer for feature learning and graph coarsening, respectively. We use a multi-layer perceptron network for classification.

### 3.4 WEIGHTED LINE GRAPH CONVOLUTION LAYER VIA INCIDENCE MATRIX

In this section, we propose to implement the WLGCL using the incidence matrix. When edge features are generated from node features as described in Section 3.3, it can significantly reduce the usage of computational resources while taking advantage of the line graph structure.

One practical challenge of using a line graph structure is that it consumes excessive computational resources in terms of memory usage and execution time. To use a line graph in a graph neural network, we need to store its adjacency matrix, compute edge features, and perform message passing operation. Our proposed WLGCL also faces this challenge. Space and time complexities of Eq. (2), which plays the dominating role, are $O(E^2) = O(N^4)$ and $O(E^2C) = O(N^4C)$, respectively. Here, we set $C_n = C_e$ for simplicity. To address this issue, we propose to use the incidence matrix $\boldsymbol{B}$ to compute the weighted line graph adjacency matrix $\boldsymbol{F}$. The adjacency matrix $\boldsymbol{F}$ can be accurately computed with the following theorem.

**Theorem 2.** *Given an undirected graph, its incidence matrix $\boldsymbol{B} \in \mathbb{R}^{N \times E}$, and its degree matrix $\boldsymbol{D} \in \mathbb{R}^{N}$, the adjacency matrix $\boldsymbol{F} \in \mathbb{R}^{E \times E}$ of the weighted line graph with weights defined by Eq. (1) can be exactly computed by*

$$\boldsymbol{F} = \boldsymbol{B}^T diag\left(\boldsymbol{D}\right)^{-1} \boldsymbol{B}, \tag{6}$$

*where $diag(\cdot)$ takes a vector as input and constructs a squared diagonal matrix using the vector elements as the main diagonal elements.*

The proof of Theorem 2 is provided in the supplementary material. Based on the results from Theorem 2, we can update the equations (Eq. (2-3)) to generate $\boldsymbol{K}_L^{(\ell)}$ in the WLGCL by replacing the adjacency matrix $\boldsymbol{F}$ with Eq. (6):

$$\begin{aligned} \boldsymbol{K}_L^{(\ell)} &= \boldsymbol{B}^{(\ell)} \boldsymbol{F}^{(\ell)} \boldsymbol{B}^{(\ell)T} \boldsymbol{X}^{(\ell)} = \boldsymbol{B}^{(\ell)} \boldsymbol{B}^{(\ell)T} \text{diag}\left(\boldsymbol{D}\right)^{-1} \boldsymbol{B}^{(\ell)} \boldsymbol{B}^{(\ell)T} \boldsymbol{X}^{(\ell)} \\ &= \boldsymbol{H}^{(\ell)} \text{diag}\left(\boldsymbol{D}\right)^{-1} \boldsymbol{H}^{(\ell)} \boldsymbol{X}^{(\ell)}, \end{aligned} \tag{7}$$

where $\boldsymbol{B}^{(\ell)T} \boldsymbol{X}^{(\ell)}$ computes edge features using node features. Notably, $\boldsymbol{H}^{(\ell)} = \boldsymbol{B}^{(\ell)} \boldsymbol{B}^{(\ell)T}$ only needs to be computed once. With computed $\boldsymbol{K}_L^{(\ell)}$, we output the new feature matrix $\boldsymbol{X}^{(\ell+1)}$ using equations Eq. (4) and Eq. (5).

By using the implementation in Eq. (7), space and time complexities of the WLGCL are reduced to $O(N \times E) = O(N^3)$ and $O(N^2 \times E) + O(N^2 \times C) = O(N^4)$, respectively. Compared to the naive WLGCL implementation, they are reduced by a factor of $N$ and $C$, respectively. In the experimental study part, we show that the WLGCL implemented as Eq. (7) dramatically saves the computational resources compared to the naive implementation. Notably, the results in Eq. (6) can be applied to other graph neural networks that leverage the benefits of line graph structures.

### 3.5 WEIGHTED LINE GRAPH CONVOLUTIONAL NETWORKS

In this section, we build a family of weighted line graph convolutional networks (WLGCNets) that utilize our proposed WLGCLs. In WLGCNets, an embedding layer such as a fully-connected layer or GCN layer is firstly used to learn low-dimensional representations for nodes in the graph. Then

Table 1: Comparison of WLGCNet and previous state-of-the-art models including WL (Shervashidze et al., 2011), PSCN (Niepert et al., 2016), DGCNN, SAGPool (Lee et al., 2019), DIFF-POOL, g-U-Net, and GIN on graph classification datasets. We report the graph classification accuracies (%) on PROTEINS, D&D, IMDB-MULTI, REDDIT-BINARY, REDDIT-MULTI5K, COLLAB, and REDDIT-MULTI12K datasets.

| | PROTEINS | D&D | IMDBM | RDTB | RDT5K | COLLAB | RDT12K |
|---|---|---|---|---|---|---|---|
| *graphs* | 1113 | 1178 | 1500 | 2000 | 4999 | 5000 | 11929 |
| *nodes* | 39.1 | 284.3 | 13 | 429.6 | 508.5 | 74.5 | 391.4 |
| *classes* | 2 | 2 | 3 | 2 | 5 | 3 | 11 |
| WL | $75.0 \pm 3.1$ | $78.3 \pm 0.6$ | $50.9 \pm 3.8$ | $81.0 \pm 3.1$ | $52.5 \pm 2.1$ | $78.9 \pm 1.9$ | $44.4 \pm 2.1$ |
| DGCNN | $75.5 \pm 0.9$ | $79.4 \pm 0.9$ | $47.8 \pm 0.9$ | - | - | $73.8 \pm 0.5$ | $41.8 \pm 0.6$ |
| PSCN | $75.9 \pm 2.8$ | $76.3 \pm 2.6$ | $45.2 \pm 2.8$ | $86.3 \pm 1.6$ | $49.1 \pm 0.7$ | $72.6 \pm 2.2$ | $41.3 \pm 0.8$ |
| DIFFPOOL | 76.3 | 80.6 | - | - | - | 75.5 | 47.1 |
| SAGPool | 71.9 | 76.5 | - | - | - | - | - |
| g-U-Net | $77.6 \pm 2.6$ | $82.4 \pm 2.9$ | $51.8 \pm 3.7$ | $85.5 \pm 1.3$ | $48.2 \pm 0.8$ | $77.5 \pm 2.1$ | $44.5 \pm 0.6$ |
| GIN | $76.2 \pm 2.8$ | $82.0 \pm 2.7$ | $52.3 \pm 2.8$ | $92.4 \pm 2.5$ | $57.5 \pm 1.5$ | $80.6 \pm 1.9$ | - |
| **WLGCNet** | $\mathbf{78.9 \pm 4.2}$ | $\mathbf{83.8 \pm 2.8}$ | $\mathbf{56.1 \pm 3.6}$ | $\mathbf{94.1 \pm 2.2}$ | $\mathbf{58.2 \pm 3.2}$ | $\mathbf{83.1 \pm 7.9}$ | $\mathbf{50.3 \pm 1.5}$ |

we stack multiple blocks, each of which consists of a WLGCL and a pooling layer (Gao & Ji, 2019). Here, the WLGCL encodes high-level features while the pooling layer outputs a coarsened graph. We use the gPool layer to produce a coarsened graph that helps to retain original graph structure information. To deal with the variety of graph sizes in terms of the number of nodes, we apply global readout operations on the outputs including maximization, averaging and summation (Xu et al., 2018). The outputs of the first GCN layer and all blocks are stacked together in the feature dimension and fed into a multi-layer perceptron network for classification. Figure 4 provides an example of our WLGCNets.

## 4 EXPERIMENTAL STUDY

In this section, we evaluate our proposed WLGCL and WLGCNet on graph classification tasks. We demonstrate the effectiveness of our methods by comparing our networks with previous state-of-the-art models in terms of the graph classification accuracy. Besides, we evaluate the efficiency of our implementation of the WLGCL in terms of the usage of computational resources. We conduct ablation experiments to demonstrate the contributions of our methods. The code and experimental setups are provided in the supplementary material.

### 4.1 PERFORMANCE STUDY

To evaluate our methods and WLGCNets, we conduct experiments on graph classification tasks using seven datasets; those are PROTEINS, D&D (Dobson & Doig, 2003), IMDB-MULTI (IMDBM), REDDIT-BINARY (RDTB), REDDIT-MULTI5K (RDT5K), COLLAB, and REDDIT-MULTI12K (RDT12K) (Yanardag & Vishwanathan, 2015). REDDIT datasets are benchmarking large graph datasets used for evaluating graph neural networks in the community. On the datasets without node features such as RDT12K, we use one-hot encodings of node degrees as node features (Xu et al., 2018). To produce less biased evaluation results, we follow the practices in (Xu et al., 2018; Zhang et al., 2018) and perform 10-fold cross-validation on training datasets. We use the average accuracy across 10 fold testing results with variances.

We report the graph classification accuracy along with performances of previous state-of-the-art models. The results are summarized in Table 1. We can observe from the results that our proposed WLGCNets significantly outperform previous models by margins of 1.3%, 1.8%, 3.8%, 1.7%, 0.7%, 2.5%, 3.2% on PROTEINS, D&D, IMDB-MULTI, REDDIT-BINARY, REDDIT-MULTI5K, COL-LAB, and REDDIT-MULTI12K datasets, respectively. The promising results, especially on large benchmarking datasets such as REDDIT-MULTI12K, demonstrate the effectiveness of our proposed methods and models for network embeddings. Note that our WLGCNet uses the gPool layer from the g-U-Net. The superior performances of WLGCNets over the g-U-Net demonstrate the performance gains are from our proposed WLGCLs.

Table 2: Comparison of WLGCL and the WLGCL using native implementation (denoted as WLGCL$_n$). We evaluate them on simulated data with different graph sizes in terms of the number of nodes and the number of edges. All layers output 64 feature channels. We report the number of multiply-adds (MAdd), the amount of memory usage, and the CPU execution time. We describe the input graph size in the format of "number of nodes / number of edges".

| Input | Operator | MAdd | Saving | Memory | Saving | Time | Speedup |
|---|---|---|---|---|---|---|---|
| 1000/50000 | WLGCL$_n$ | 166.47B | 0.00% | 9.5GB | 0.00% | 15.73s | 1.0× |
| | WLGCL | 51.13B | 69.28% | 0.19GB | 97.94% | 0.63s | 26.2× |
| 1000/100000 | WLGCL$_n$ | 652.87B | 0.00% | 37.63GB | 0.00% | 56.62s | 1.0× |
| | WLGCL | 101.13B | 84.51% | 0.38GB | 98.98% | 1.21s | 47.2× |
| 1000/150000 | WLGCL$_n$ | 1,459.27B | 0.00% | 86.71GB | 0.00% | 134.36s | 1.0× |
| | WLGCL | 151.13B | 89.64% | 0.57GB | 99.34% | 1.82s | 74.6× |
| 2000/150000 | WLGCL$_n$ | 1,478.66B | 0.00% | 99.87GB | 0.00% | 278.8s | 1.0× |
| | WLGCL | 608.52B | 58.85% | 1.13GB | 98.86% | 6.19s | 45.1× |

## 4.2 COMPUTATIONAL EFFICIENCY STUDY

In Section 3.4, we propose an efficient implementation of WLGCL using the incidence matrix, which can save dramatic computational resources compared to the naive one. Here, we conduct experiments on simulated data to evaluate the efficiency of our methods. We build networks that contain a single layer to remove the influence of other factors. We conduct experiments on graphs of different sizes in terms of the number of nodes. Since WLGCL takes advantage of line graph structures, the graph density has a significant impact on the layer efficiency. Here, the graph density is defined as $2E/(N \times (N-1))$. To investigate the impact of the graph density, we conduct experiments on graphs with the same size but different numbers of edges.

By using the TensorFlow profile tool (Abadi et al., 2016), we report the computational resources used by networks including the number of multiply-adds (MAdd), the amount of memory usage, and the CPU execution time. The comparison results are summarized in Table 2. We can observe from the results that the WLGCLs with our proposed implementation use significantly less computational resources than WLGCLs with naive implementation in terms of the memory usage and CPU execution time. By comparing the results on first three inputs, the advantage on efficiency of our method over the naive implementation becomes much larger as the increase of the graph density with the same graph size. When comparing results of the last two inputs with the same number of edges but different graph sizes, we can observe that the efficiency advantage of our proposed methods remains the same. This shows that the graph density is a key factor that influences the usage of computational resources, especially on dense graphs.

## 4.3 RESULTS ON SMALL DATASETS

In the previous section, we evaluate our methods on benchmarking datasets that are relatively large in terms of the number of graphs and the number of nodes in graphs. To provide a comprehensive evaluation, we conduct experiments on relatively small datasets to evaluate the risk of over-fitting of our methods. Here, we use three datasets; those are MUTAG (Wale et al., 2008), PTC (Toivonen et al., 2003), and IMDB-BINARY (Borgwardt et al., 2005). MUTAG and PTC datasets are bioinformatics datasets with categorical features on nodes. We follow the same experimental settings as in Section 4.1. The results in terms of the graph clas-

Table 3: Comparison of WLGCNet and previous state-of-the-art models on relatively small datasets. We report the graph classification accuracies on MUTAG, PTC, and IMDBB datasets.

| | MUTAG | PTC | IMDBB |
|---|---|---|---|
| *graphs* | 188 | 344 | 1000 |
| *nodes* | 17.9 | 25.5 | 19.8 |
| WL | 90.4 ± 5.7 | 59.9 ± 4.3 | 73.8 ± 3.9 |
| DGCNN | 85.8 ± 1.7 | 58.6 ± 2.4 | 70.0 ± 0.9 |
| PSCN | 92.6 ± 4.2 | 60.0 ± 4.8 | 71.0 ± 2.2 |
| g-U-Net | 87.2 ± 7.8 | 64.7 ± 6.8 | 75.4 ± 3.0 |
| GIN | 90.0 ± 8.8 | 64.6 ± 7.0 | 75.1 ± 5.1 |
| **WLGCNet** | **93.0 ± 5.8** | **72.7 ± 6.0** | **78.8 ± 5.1** |

sification accuracy are summarized in Table 3 with performances of previous state-of-the-art models. We can observe from the results that our WLGCNet outperforms previous models by margins of 0.4%, 6.0%, and 3.4% on MUTAG, PTC, and IMDB-BINARY, respectively. This demonstrates that our proposed models will not increase the risk of the over-fitting problem even on small datasets.

## 4.4 ABLATION STUDY OF WEIGHTED LINE GRAPH CONVOLUTION LAYERS

In this section, we conduct ablation studies based on WLGCNets to demonstrate the contribution of our WLGCLs to the entire network. To explore the advantage of line graph structures, we construct a network that removes all layers using line graphs. Based on the WL-GCNet, we replace WLGCLs by GCNs using the same number of trainable parameters, which we denote as $\text{WLGCNet}_g$. To compare our weighted line graph with the regular line graph, we modify our WLGCLs to use regular line graph structures. We denote the resulting network as $\text{WLGCNet}_l$. We evaluate

Table 4: Comparison of WLGCNet, the network using the same architecture as WLGCNet with GCN layers (denoted as $\text{WLGCNet}_g$), the network using the same architecture as WLGC-Net with regular line graph convolution layers (denoted as $\text{WLGCNet}_l$). We report the graph classification accuracies on REDDIT-BINARY, REDDIT-MULTI5K, and REDDIT-MULTI12K.

|  | RDTB | RDT5K | RDT12K |
|---|---|---|---|
| $\text{WLGCNet}_g$ | $93.2 \pm 1.5$ | $56.9 \pm 2.2$ | $49.1 \pm 1.5$ |
| $\text{WLGCNet}_l$ | $93.6 \pm 2.0$ | $57.3 \pm 3.0$ | $49.6 \pm 2.8$ |
| WLGCNet | $\mathbf{94.1 \pm 2.2}$ | $\mathbf{58.2 \pm 3.2}$ | $\mathbf{50.3 \pm 1.5}$ |

these networks on three datasets; those are REDDIT-BINARY, REDDIT-MULTI5K, and REDDIT-MULTI12K datasets. Table 4 summaries the graph classification results. We can observe from the results that both WLGCNet and $\text{WLGCNet}_l$ achieve better performances than $\text{WLGCNet}_g$, which demonstrates the benefits of utilizing line graph structures on graph neural networks. When comparing WLGCNet with $\text{WLGCNet}_l$, WLGCNet outperforms $\text{WLGCNet}_l$ by margins of 0.5%, 0.5%, and 0.7% on REDDIT-BINARY, REDDIT-MULTI5K, and REDDIT-MULTI12K datasets, respectively. This indicates that our proposed WLGCL utilizes weighted line graph structures with unbiased topology information encoded, thereby leading to better performances.

## 4.5 NETWORK DEPTH STUDY

Network depth in terms of the number of blocks is an important hyper-parameter in the WLGC-Net. In previous experiments, we use three blocks in WLGCNets based on our empirical experiences. In this section, we investigate the impact of the network depth in WLGCNets on network embeddings. Based on our WLGCNet, we vary the network depth from 1 to 5, which covers a reasonable range. We evaluate these networks on PTC, PROTEINS, and REDDIT-BINARY datasets and report the graph classification accuracies. Figure 5 plots the results of WLGCNets with different numbers of blocks. We can observe from the figure that the best performances are achieved on WLGCNets with three blocks on all three datasets. When the network depth increases, the performances decrease, which indicates the over-fitting issue.

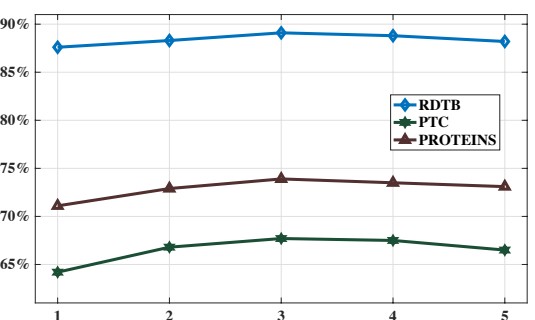

Figure 5: Comparison of WLGCNets with different depths on PTC, PROTEINS, and REDDIT-BINARY. We report the classification accuracies.

## 5 CONCLUSION

In this work, we consider the biased topology information encoding in graph neural networks that utilize line graph structures to enhance network embeddings. A line graph constructed from a graph can encode the topology information. However, the dynamics in the line graph are inconsistent with that in the original graph. On line graphs, the features of nodes with high degrees are more frequently passed in the graph, which causes understatement or overstatement of node features. To address this issue, we propose the weighted line graph that assigns normalized weights on edges such that the weighted degree of each node is 2. Based on the weighted line graph, we propose the weighted line graph layer that leverages the advantage of the weighted line graph structure. A practical challenge faced by graph neural networks on line graphs is that they consume excessive computational resources, especially on dense graphs. To address this limitation, we propose to use the incidence matrix to implement the WLGCL, which can dramatically save the computational resources. Based on the WLGCL, we build a family of weighted line graph convolutional networks.

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

## A    EXPERIMENTAL SETUP

We describe the experimental setup for graph classification tasks. In this work, we mainly evaluate our methods on graph classification datasets such as social network datasets and bioinformatics datasets. The node features are created using one-hot encodings and fed into the networks. In WLGCNets, we use GCN layers as the graph embedding layers. After the first GCN layer, we stack three blocks as described in Section 3.5. The outputs of the GCN layer and WLGCLs in three blocks are processed by a readout function and concatenated as the network output. The readout function performs three global pooling operations; those are maximization, averaging, and summation. The network outputs are fed into a classifier to produce predictions. Here, we use a two-layer feed-forward network with 512 units in the hidden layer as the classifier. We apply dropout (Srivastava et al., 2014) on the network and the classifier.

We use an Adam optimizer (Kingma & Ba, 2015) with a learning rate of 0.001 to train WLGCNets. To prevent over-fitting, we apply the $L_2$ regularization on trainable parameters with a weight decay rate of 0.0008. All models are trained for 200 epochs using one NVIDIA GeForce RTX 2080 Ti GPU on an Ubuntu 18.04 system.

## B    PROOF FOR THEOREM 1

*Proof.* Given nodes $a$ and $b$ with degrees $D_a$ and $D_b$ in a graph $\mathbb{G}$, a node $(a, b)$ in the corresponding weighted line graph $WL(\mathbb{G})$ connects to $D_a - 1$ and $D_b - 1$ nodes through $a$ and $b$ in $\mathbb{G}$, respectively. The weighted degree of the node $(a, b)$ is computed by summing up the weights of edges that connect $(a, b)$ to other nodes through $a$ and $b$, and the weight of its self-loop:

$$
\begin{aligned}
WLD_{(a,b)} &= \sum_{i=1}^{D_a-1} \frac{1}{D_a} + \sum_{j=1}^{D_b-1} \frac{1}{D_b} + \left( \frac{1}{D_a} + \frac{1}{D_b} \right) \\
&= \sum_{i=1}^{D_a} \frac{1}{D_a} + \sum_{j=1}^{D_b} \frac{1}{D_b} = 2.
\end{aligned}
\tag{8}
$$

This completes the proof. □

## C    PROOF FOR THEOREM 2

*Proof.* We construct a weighted incidence matrix by normalizing the weights as $\hat{B}_{i,(i,j)} = 1/D_i$. Thus, the weighted incidence matrix is computed as $\hat{B} = \text{diag}(D)^{-1} B$. In the incidence graph, each edge is connected to its two end nodes. Thus, each column in the incidence matrix $B_{:,(a,b)}$ has two non-zero entries; those are $B_{a,(a,b)}$ and $B_{b,(a,b)}$. The same rule applies to the weighted incidence matrix $\hat{B}$. Based on this observation, we have

$$
\begin{aligned}
\left( B^T \hat{B} \right)_{(a,b),(b,c)} &= \sum_{i=1}^{N} B^T_{(a,b),i} \times \hat{B}_{i,(b,c)} \\
&= B^T_{(a,b),a} \hat{B}_{a,(b,c)} + B^T_{(a,b),b} \hat{B}_{b,(b,c)} \\
&= \begin{cases} \frac{1}{D_b} & \text{if } a \neq c \\ \frac{1}{D_b} + \frac{1}{D_a} & \text{if } a = c \end{cases} = F_{(a,b),(b,c)}.
\end{aligned}
\tag{9}
$$

This completes the proof. □

