# OpenReview forum: "Weighted Line Graph Convolutional Networks"
_ICLR.cc/2021/Conference — Reject_

### Official Review · AnonReviewer1 · 2020-10-26
**Weighted Line Graph Convolutional Networks**

**Rating:** 5
**Confidence:** 4

**Review:**

The paper introduces a GNN architecture that is based on a weighted line-graph transformation (in conjunction with a node-based architecture).
The central idea is that a normal line graph transformation will lead to an over-representation of certain nodes (motifs) in the network, which the authors strive to correct.
The efficacy of these ideas is tested in a range of numerical experiments.

Pro:
* The presentation is quite clear, in my opinion apart from some smaller inaccuracies.
* The experiments constructed appear thorough and rigorous.

Cons:
* A more precise theoretical justification why their algorithm should work better (has more expressive power than other architectures) is missing
* The improvements seen are not strong compared to other architectures
* It seems that most of the computation could be implemented w/o ever appealing to a line graph transformation.

Additional comments:
The general idea of the paper sounds appealing. A line graph may be seen as a "higher-order" representation of a graph and thus we may (intuitively) hope that there is more information to be gained here.
However, the authors argument hinges upon a single paper that shows some improvements for community detection using line-graphs, but not much further theoretical justification.
The fact that high-degree nodes will be over-represented (as they will have more edges in the line-graph), does not necessary mean that normalization will help. The neural network could also learn to take into account this information, in principle?
In particular, we know that GNN cannot be more expressive that the Weisfeiler Leman algorithm (see e.g., Morris et al); does a line-graph transformation help here and make the architecture more akin to a 2-WL network?
I feel that some kind of stronger theoretical justification is missing here, given the computational costs of expanding the representation to edges.
The numerical results are not as strong to convince me in this regard.

The two theorems proven are very basic linear algebra, and cannot really be counted as a theoretical contribution here, in my opinion.
In particular all these ideas have already been presented in a lot of detail in the work by Evans and Lambiotte. A second citation that is missing in this context in my opinion is:
Evans, Tim S., and Renaud Lambiotte. "Line graphs of weighted networks for overlapping communities." The European Physical Journal B 77.2 (2010): 265-272.

Looking at equation (7) we see that essentially all the computations in layer \ell can be done  using node x node matrices (H, D, X), i.e., the implementation does not even need to appeal to any kind of line-graph transformation. This makes the rationale that the line-graph representation improves the learning somewhat questionable --- after all there is just a message passing between nodes going on here.

In summary, the paper provides a mixture of theory and experiments.
Unfortunately, I feel that both the theory part and the experiments are not fully satisfactory. The theory part does not provide clear theoretical guarantees or insights why the proposed architectures would perform better, in general, I think. Moreover the computations still seem to boil down to a node-to-node message passing.
If the authors could provide a more rigorous justification here, this would significantly improve the paper.

The experiments appear to be rigorous, but the results are essentially on par with a number of other fully "node-based" methods, so it is not fully clear whether there is really a clear advantage here.

Minor comments:
How is the incidence matrix B defined? It seems an unsigned version is used, which would be less common in a graph / network theory.

---

> ### Author Response · Authors · 2020-11-24
> **Rebuttal to AnonReviewer1**
>
> Question 1: Theoretical justification of using line graph structure.
>
> Answer: The advantage of line graph structure has been explored on various graph embedding tasks such as [1, 2, 3]. Essentially, line graph structure can provide a different perspective to encode network embeddings. Also, it will give a good emphasize on edge features in graph neural networks.
>
> Question 2: equation (7) we see that essentially all the computations in layer \ell can be done using node x node matrices.
>
> Answer: There may be a misunderstanding here. Equation (7) provides an efficient computation when node features are utilized. However, when there are edge features, which is an important motivation of using line graph structure, we need to use the computations (2-5) in Section 3.3.
>
> Question 3: results are essentially on par with a number of other fully "node-based" methods
>
> Answer: There may be a misunderstanding here. We mainly compare our methods with previous state-of-the-art models such as DGCNN, DIFFPOOL, SAGPool, g-U-Net, and GIN on graph classification tasks in Tables 1 and 3. Compared with previous methods, our methods can outperform them by 1% to 3% on most benchmarking datasets. Given that these datasets are popular benchmarking datasets and heavily explored, we consider these improvements to be significant and they can demonstrate the effectiveness of our methods.

---

### Official Review · AnonReviewer2 · 2020-10-27
**Interesting paper but some mathematical details are not clearly presented**

**Rating:** 6
**Confidence:** 4

**Review:**

This paper proposes passing messages on the line graph for learning representations of graphs. To overcome the bias that high-degree nodes are over-emphasized during message passing, the authors propose to reweight edges in the line graph. Then it performs message passing on both the original graph and the weighted line graph. The authors also use incidence matrix in their model to reduce computation overhead. Overall, this paper is well written and easy to follow. The experimental results demonstrate the effectiveness and efficiency of the proposed method. However, I have the following concerns:

1. In eq.1, the weight of self loops is 1/D_b + 1/D_a, but it is unclear why it should be designed like this. Does it mean that self loops are more important in the line graph? It seems more likely for the purpose of deriving Theorem 1.

2. In eq. 2-5, how is the edge feature matrix Y_l updated to Y_{l+1} is not specified.

3. In experiments the authors only conduct experiments on graph classification. Can the proposed method be applied to node classification? Does it work on large graphs with thousands of nodes?

---

> ### Author Response · Authors · 2020-11-24
> **Rebuttal to AnonReviewer2**
>
> Question 1: In eq.1, the weight of self-loops is 1/D_b + 1/D_a, but it is unclear why it should be designed like this.
>
> Answer: Yes. The weights designed like this will emphasize on the self-loops, which consequently results in the emphasis of node’ own features. In regular deep neural networks, skip connections are commonly used, which can also emphasize own features. Thus, we believe such design can reasonably emphasize nodes’ features.
>
>  Question 2: In eq. 2-5, how is the edge feature matrix Y_l updated to Y_{l+1} is not specified.
>
> Answer: Thanks for pointing out this. Yes, we didn’t provide an equation for edge feature updates here. Actually, the update rule is specified in the sentence below Equation (7):
>
> “B^T X computes edge features using node features”.
>
> We will add this update rule in to eq(2-5) to make it clear and complete.
>
> Question 3: Application to node classification tasks
>
> Answer: Yes, our methods can be applied to node classification tasks by removing the readout layer from WLGCNets. We will add more results on node classification tasks in the final version.

---

### Official Review · AnonReviewer4 · 2020-10-28
**Overall, I vote to reject the paper. I do not disagree with the main premise of the paper but feel that it needs some rewriting as well as additional empirical evaluation.**

**Rating:** 4
**Confidence:** 3

**Review:**

# Summary
This paper introduces a weighted line graph formulation (WLGCL) which corrects the over-counting ("bias") of high-degree node features in a line-graph based convolutional network. Further, the paper uses Incidence Matrix to implement WLGCL updates which reduces the space complexity ($O(N^4) \to O(N^3)$) and time complexity ($O(N^4 C) \to O(N^4)$) compared to the naive implementation. The paper shows empirical evaluation on downstream task of graph classification and shows gain in accuracy.

# Observations
- The use of the word "dynamics" (Page 1, paragraph 2) in the context of over-representation of node features in the message passing is very confusing. Typically, "dynamics" in the context of graph networks often implies changing graph structure which is not the case here. The usage is taken from [Evans & Lambiote, 2009] (https://arxiv.org/pdf/0903.2181.pdf) but in that context it refers to random walks.

- The paper mentions on Page 4 last line that "advanced feature aggregation methods such as GAT" can easily be applied to the line graph. This should be demonstrated, e.g., in the supplementary material.  As a note, in the paper [Monti et al. 2018] (available only on https://arxiv.org/pdf/1806.00770.pdf) also presents line-graph formulation with GAT applied to the line graph.

- [Bandhopadhyay et al. 2019]  also present a weighted line graph but since it is only present on Arxiv (https://arxiv.org/abs/1912.05140), I disregard lack of comparison with that paper.

- It is not clear which datasets have node features - this should be clearly mentioned preferably in the main text but if not definitely in the supplementary material.

-  The paper claims performance improvement over graph U-net which demonstrates the benefit of weighted approach which does unbiased node-feature updates. For PROTEINS and COLLAB datasets, the standard deviation of the proposed method is significantly higher than competing methods - what would be the possible explanation for this?

- The benefit of using Incidence Matrix based updates is clear in terms of space and time complexity. However, I would like to see comparison with other methods on space/time complexity if focussing solely on graph classification task.

# Optional remarks
- It is not necessary but might be useful to show evaluation on other downstream tasks.  For example, in downstream task of node classification - CORA, Citeseer, Pubmed (see e.g., the Graph U-nets paper).

- The node feature bias might be more relevant when there are edge features as well. To my understanding, the current datasets do not have edge features.  For example, please see the CensNet paper (cited as Jiang, 2019 in this paper) and the multi-task classification datasets (_Tox21_ and _Lipophilicity_)

# Minor comments
- What is the dropout rate (Appendix A)?
- As the model trains, a plot showing the test metrics would be good in the supplementary.

# Recommendation
Overall, I vote to reject the paper. I do not disagree with the main premise of the paper but feel that it needs some rewriting as well as additional empirical evaluation.

---

> ### Author Response · Authors · 2020-11-24
> **Rebuttal to AnonReviewer4**
>
> Question 1: The use of the word "dynamics" is confusing.
>
> Answer: Thank you for pointing out this. We will fix this word to reduce the confusion.
>
> Question 2: It is not clear which datasets have node features.
>
> Answer: Thank you for pointing out this. Social network datasets including COLLAB, IMDB, and REDDIT don’t have node features while bioinformatic datasets such as MUTAG, PTC, PROTEINS, and D&D have node features. We will clarify this in the final version.
>
> Question 3: Possible explanation for higher standard deviation on PROTEINS and COLLAB datasets.
>
> Answer: Since these datasets are relatively small, the original partition for 10-fold cross validation may generate folds with biased training examples distribution. It would be improved if a better partition scheme can be used. However, to ensure fair comparison, we strictly follow the same experimental settings as Graph U-Net and GIN.
>
> Question 4: I would like to see comparison with other methods on space/time complexity
>
> Answer: In the paper, we mainly compare our methods with the original line graph implementation to improve efficiency. However, it is still hard to compete with other methods that utilize original graph structures. It can be considered as a trade-off between performance and efficiency. Using line graph structure can improve the performance but also involve efficiency issues. Our proposed implementation methods aim to provide a better trade-off between them.

---

### Official Review · AnonReviewer3 · 2020-10-29
**Thorough paper on propagating over line graphs**

**Rating:** 6
**Confidence:** 4

**Review:**

Summary of contribution:
This paper discusses graph convolutional networks.  The authors recap existing work to observe that models already exist to propagate information through graphs in two different ways:
1. Standard propagation passes information from nodes along edges to neighboring nodes
2. Line graph propagation adopts the same scheme on the *line graph* of the original graph:  original edges become vertices in the line graph, and these new vertices are connected when they share an endpoint in the original graph.

The authors suggest a model that simultaneously propagates information in both graphs, coupling the two propagations at each step.

They additionally observe that high-degree vertices in the original graph induce cliques in the line graph, which will naively result in over-weighting such vertices, as they have many more paths along which to propagate information.  They propose reweighting the line graph so that vertices in the line graph all have weighted degree 2.

They perform a series of experiments using this combined propagation scheme through the graph and the weighted line graph.  The experiments cover graph classification tasks, and are well thought out.  There are a number of tasks with larger graphs, plus experiments with smaller graphs to test over-fitting.  There are a few ablation studies, and a sensitivity study on depth of their architecture.

Strong points:
* The weighting argument for line graphs seems natural, and as far as I can tell the authors are the first to propose this.
* The coupling of propagation in the two graphs is elegantly established and also seems natural
* The experiments are well-defined, and the results themselves are pretty compelling (but please see question below)

Weak points / concerns:
* The main proposed points of novelty in the paper are as follows, as far as I can tell:
  N1: Weighted line graphs.  Line graphs themselves have been used before, and the weighting scheme itself is straightforward -- I believe it has been used before in combinatorial algorithms based on line graphs.
  N2: There is a small optimization to avoid materializing the cliques in the line graph during propagation, but this also seems natural; in fact, it would be a bug not to retain the propagation structure in this clearly more efficient representation.
  N3: The coupling of propagations in equation 5 seems nice.
These haven't been studied before, and N3 seems quite nice, but I don't see significant mathematical advances in this structure or significant advances in information flow.  I still feel pretty good about the contributions of the paper as they combine a number of natural steps, and they show good empirical outcomes.

Questions for the authors:
Q1. In Table 1, I'm surprised that the WLGCNet has confidence intervals that overlap significantly with, say, GIN for many of the datasets.  However, there is never an inversion in the order of the winning system.  Depending on how the CIs were computed, this seems like an unexpected coincidence, which makes me think I may not understand the computation of the confidence intervals here.  Would appreciate some clarification.
Q2. The optimization holds only for graphs where the edge features are materialized from the vertex features.  Does this hold for all the datasets?

---

> ### Author Response · Authors · 2020-11-24
> **Rebuttal to AnonReviewer3**
>
> We really appreciate your good words and valuable suggestions.
>
> Question 1: clarification on results in Table 1.
>
> Answer: In our experiments, we strictly follow the same settings as GIN. In particular, we perform 10-fold cross-validation on training datasets since most of benchmarking datasets are very small. The reported confidence intervals are computed based on the accuracy results of 10 folds.
>
> Question 2: The optimization holds only for graphs where the edge features are materialized from the vertex features. Does this hold for all the datasets?
>
> Answer: Yes, you are correct. We will continue to explore efficient implementation such that our methods can benefit datasets with edge features.

---

### Official Review · AnonReviewer5 · 2020-11-07
**Review for "Weighted Line Graph Convolutional Networks"**

**Rating:** 5
**Confidence:** 4

**Review:**

The paper proposed a GNN model based on a weighted line graph, which adds weights to the line graph for the original input graph in a node/graph property prediction task. The line graph is a graph built on the original graph but with edges as nodes. A new convolution called weighted line graph convolution layer (WLGCL) is proposed to overcome the issue of "biased topological information" of the line graph. The weights for the line graph in WLGCL are computed based on the node degree of the original graph, which implies the node degree in the line graph is always 2. The WLGCL can be implemented for different kinds of graph convolution, which rule incorporates graph connectivity, node features and edge features.
Experiments compared the performance of the proposed model with existing GNN methods on graph classification tasks and computational complexity with other methods.
1. The WLGCL introduces the weights for edges in line graph convolution, which reduces the computational cost. The performance of WLGCL on some graph classification datasets are good.
2. The WLGCL is a weighted version of the line graph neural networks (LGNNs) as studied previously in [Chen, Li, Bruna, Supervised Community Detection with Line Graph Neural Networks, ICLR 2019].
Besides saving the computational cost and removing biased degree information, what are other benefits over LGNN? Is there a significant improvement of the test accuracy against LGNN on various types of graph datasets? Maybe saying “biased topological information” here is misleading as what change the WLGCL makes compared to LGNN is the node degree.
3. The experiments, like Table 1, compare with some existing GNNs methods. The author should compare with more existing GNNs, such as GAT. The new datasets, Open Graph Benchmarks, should also be tested to show the performance of the proposed GNN model.
4. What is the performance of WLGCL with the normalisation of the adjacency matrix on graph classification tasks?
5. The study of the test accuracy vs depth of the GNN with WLGCL indicates the WLGCL may work in deep nets. Will increase the depth further be beneficial or not? Is there any interpretation from information theory?

---

> ### Author Response · Authors · 2020-11-24
> **Rebuttal to AnonReviewer5**
>
> Question 2: Comparison with LGCNN
>
> Answer: We have made comparisons with the regular line graph (LGNN) on various datasets including REDDIT-BINARY, REDDIT-MULTI5K, and REDDIT-MULTI12K datasets. The comparison results are illustrated in table 4. In particular, our WLGCN outperforms LGCN by 0.5%, 0.9%, and 0.7% on REDDIT-BINARY, REDDIT-MULTI5K, and REDDIT-MULTI12K datasets, respectively. These results demonstrate the advantage of our WLGCN over LGCN.
>
> Question 3: Comparison with GCNs and GATs
>
> Answer: We provide the comparison results with GCN on image classification datasets as in table 4. In the final version, we plan to add more comparisons on recommended datasets.
>
> Question 4: Performances with normalized adjacency matrix
>
> Answer: The results shown in tables 1, 3, and 4 are performances of using normalized adjacency matrix. We only do simple adjacency matrix normalization. In particular, we normalize each node by its incoming degree such that the aggregated features are scaled.
>
> Question 5: Performance of deep WLGCNets
>
> Answer: The performances of WLGCNets achieves the best performance when the depth of networks is 3. We can observe from Figure 5 that when the depth increases, the performances decrease due to over-fitting issues.

---

### Decision · Program_Chairs · 2021-01-07
**Final Decision**

**Decision:**

Reject

**Comment:**

The paper proposed a GNN model based on a weighted line graph (dual of the input graph), where information is simultaneously propagated on both graphs, coupling the two propagations at each step.

Overall, the reviewers were lukewarm about the paper, with some raised criticism including
- limited novelty in light of Monti et al. 2018
- limited theoretical justification
- unconvincing and incomplete experiments, not offering significant improvement compared to other alternatives

While the presented approach is interesting, we believe the paper is below the bar and recommend Rejection.